# Somatic Mutations in Fruit Trees: Causes, Detection Methods, and Molecular Mechanisms

**DOI:** 10.3390/plants12061316

**Published:** 2023-03-14

**Authors:** Seunghyun Ban, Je Hyeong Jung

**Affiliations:** Smart Farm Research Center, Korea Institute of Science and Technology (KIST), Gangneung 25451, Gangwon, Republic of Korea; banting@kist.re.kr

**Keywords:** somatic mutation, bud sports, fruit crops, molecular mechanisms, apple, grape, citrus, peach

## Abstract

Somatic mutations are genetic changes that occur in non-reproductive cells. In fruit trees, such as apple, grape, orange, and peach, somatic mutations are typically observed as “bud sports” that remain stable during vegetative propagation. Bud sports exhibit various horticulturally important traits that differ from those of their parent plants. Somatic mutations are caused by internal factors, such as DNA replication error, DNA repair error, transposable elements, and deletion, and external factors, such as strong ultraviolet radiation, high temperature, and water availability. There are several methods for detecting somatic mutations, including cytogenetic analysis, and molecular techniques, such as PCR-based methods, DNA sequencing, and epigenomic profiling. Each method has its advantages and limitations, and the choice of method depends on the research question and the available resources. The purpose of this review is to provide a comprehensive understanding of the factors that cause somatic mutations, techniques used to identify them, and underlying molecular mechanisms. Furthermore, we present several case studies that demonstrate how somatic mutation research can be leveraged to discover novel genetic variations. Overall, considering the diverse academic and practical value of somatic mutations in fruit crops, especially those that require lengthy breeding efforts, related research is expected to become more active.

## 1. Introduction

Somatic mutation is a natural process that occurs in both plants and animals and leads to genetic diversity [1]. Unlike mutations that occur in gametes and are passed on to the next generation, somatic mutations occur in somatic cells and are usually not passed on to offspring. However, somatic mutations in plants can still potentially affect the reproductive organs of a plant, even though the mutation originally occurred in non-reproductive cells. This is because the mutated cells may still have the ability to develop into buds or flowers and contribute to the plant’s reproductive process. Especially for fruit crops, somatic mutations can lead to the emergence of novel traits, such as alterations in fruit size [2,3], color [4,5,6], or taste [7,8], which could be commercially valuable. In orchards, lateral shoots, floral clusters, or individual flowers/fruits on trees can be found with noticeably different phenotypes from the rest of the plant [9]. These are called “bud sports”, and enable the propagation of new mutations through asexual propagation while maintaining the name recognition of the existing variety and potentially gaining a market advantage with novel or improved characteristics added to the existing ones. According to Okie [10], more than 170 commercially available varieties of peaches and nectarines are derived from bud-sport mutations. At least 38 bud-sport mutations from ‘Gala’ have been reported since the original release of the ‘Gala’ apple cultivar in 1965 [11], whereas 91 bud-sport mutations have been reported from ‘Fuji’ [12]. 

Bud sports, clones, and crosses are all important for plant genetic diversity in horticulture and agriculture [13]. Clones are created through asexual propagation, where genetically identical plants are produced from a single parent plant through methods such as rooting, cuttings, or grafting. Grafting is a major propagation method for fruit trees. In contrast, crosses are formed by a sexual reproduction process that involves the crossing of two genetically different plants to produce offspring with new genetic combinations. Bud sports are genetic mutations that occur in a single bud or shoot of a plant, leading to a branch or portion of the plant with different characteristics. Each of these methods can contribute to plant genetic diversity in different ways. Understanding how somatic mutations arise and how they interact with other genetic processes can reveal the evolutionary history of fruit crops and clarify their connections to other plant species. In addition, somatic mutants are known to have a nearly identical genetic background to that of the original cultivars [14,15], making them an important source of material for research in plant genetics and physiology to understand the molecular mechanisms responsible for the variation in traits of interest.

This review aims to provide a comprehensive understanding of somatic mutations in fruit crops, covering their definitions, types, mechanisms, and genetic causes. We also explore recent studies and progress in somatic mutation research for diverse fruit crops, such as apples, grapes, and citrus. Additionally, this review paper identifies the challenges and future directions for research on somatic mutations in fruit crops.

## 2. Factors Affecting Somatic Mutation in Fruit Crops

Depending on the plant, the frequency and type of somatic mutations that occur may differ based on differences in genetic background [16]. For example, some plants may have higher natural mutation rates due to the presence and action of transposable elements, whereas others may have lower mutation rates due to DNA repair mechanisms [17]. Somatic mutations are caused by both internal and external factors (Figure 1).

Examples of internal factors include DNA replication errors [18], DNA repair errors [19], transposable elements [20,21], and deletion [22]. Environmental factors can also affect the frequency and type of somatic mutations [23]. For example, the continuous exposure of plants to mutagens in the environment, such as strong ultraviolet radiation [24], increases the likelihood of somatic mutations [25]. In addition, high temperature [26,27], salt stress [28], water availability [29], and other environmental factors can induce somatic mutations. The induction and types of somatic mutations can also be influenced by the physiological conditions of plants [30]. For example, somatic mutations are more likely to occur in actively growing tissues, such as meristems, than in mature tissues. One of the reasons for this is that the activity of transposons varies depending on the developmental stage of the plant [20,21]. Epigenetic factors can influence the stability of plant genomes and their susceptibility to mutagens [31,32]. For example, alterations in DNA methylation patterns can affect the expression of genes involved in cell-cycle regulation and DNA repair, which can consequently influence the frequency of somatic mutations. According to a recent study by Monroe et al., the frequency of mutations in Arabidopsis is lower in functional and essential genes, and more than 90% of genome-wide patterns of mutation bias are explained by epigenomic and physical features [33].

## 3. Methods for Detecting Somatic Mutation

The various methods for verifying that putative somatic mutations show characteristics different from those of the original cultivar discovered by a breeder in an orchard are shown in Table 1.

Cytogenetic techniques are used to examine the number, structure, and behavior of chromosomes to identify abnormalities such as aneuploidy, polyploidy, deletions, and translocations [34]. Flow cytometry can be used in a similar manner to cytogenetic techniques to examine chromosomes by measuring the amount of DNA in each cell and identifying aneuploidy and polyploidy, as well as changes in DNA content within individual cells [35]. DNA fingerprinting analyzes DNA fragments to distinguish genetic variations between individuals. Some examples of DNA fingerprinting techniques used for this purpose are restriction fragment length polymorphism (RFLP) [36], amplified fragment length polymorphism (AFLP) [37], and simple sequence repeats (SSR) analysis [38]. Microarray analysis can compare the levels of gene expression between somatic mutations and their original variety, enabling the identification of candidate genes for large-scale somatic mutation screening in a cost-effective and efficient manner [39,40]. RNA-Seq can also compare gene expression levels at the transcriptome level, which can be used to select candidate gene regions for somatic mutation [41,42,43]. If certain genes are not expressed in mutants, it is possible to conduct additional experiments, such as comparing the nucleotide sequences of the targeted genomic DNA regions. Recent advances in next-generation sequencing (NGS) technology have enabled the detection of whole-genome sequences in an individual laboratory, allowing for the identification of somatic mutations by comparing the DNA sequence of a mutant sample to that of the original variety [3,44].

## 4. Research on Somatic Mutation in Fruit Crops

We have summarized 36 studies (13 on apple, 9 on grape, 4 on citrus, 4 on pear, 3 on peach, 2 on orange, and 1 on plum) that characterized the somatic mutations in fruit crops (Table 2).

Regardless of the plant species, the most frequent type of research has been related to fruit color. This is because somatic mutants that differ from the original strain in fruit color are easier to detect than other types of somatic mutants. Additionally, fruit color has a significant impact on consumer preference [73,74,75], which likely explains why researchers have shown particular interest in elucidating the underlying mechanisms of these anthocyanin variants. Research on somatic mutations related to fruit maturation and ripening has been actively conducted in apples [22,44,47], citrus [56], grapes [61], peaches [66], and plums [72]. This is because the selling time of a crop variety is determined by its maturation date, and distinct storage techniques and durations are necessary based on the maturation date and ripening characteristics. Therefore, comprehensive studies on fruit maturation and ripening mechanisms have been conducted for most fruit crops. 

In addition to studies on the overall common interests of fruit crops, studies on crop-specific traits have also been actively conducted. The seedless characteristics of grapes and apples are particularly important for consumers when purchasing fruit [76]; therefore, researchers have attempted to understand the molecular mechanism using seedless somatic mutants [49,63,64]. Research on apple tree morphology is a representative example of crop-specific research. In the 1960s, a bud mutation called ‘Wijcik McIntosh’ was discovered in McIntosh apples that exhibited vertically growing branches and a reduced number of branches [77]. This trait has received continued interest from the apple community as it requires less pruning and is suitable for mechanical fruit harvesting; this has led to active research and the discovery of its molecular mechanism [50,51,52,53]. In the case of peach fruit, the typical shape is round; however, since the discovery of flat peach fruit [78,79], the associated molecular mechanisms have been actively researched. Fruit shape in peaches has a significant impact on consumer preference, and as the popularity of flat fruits increases, these studies help increase the understanding of fruit shape [67,68,80].

## 5. Challenges and Future Direction

Numerous studies have been conducted on somatic mutations; however, these studies have various challenges and limitations. One of the main challenges is to distinguish between true mutations and genetic variations. For example, differences in transcription-factor genes such as *MYB1* [4,5,59,60], *MYB10* [6,70,71], and *MYB90* [46] cause different coloration in mutants compared to the original variety. Although differences in the expression and methylation of *MYB* could explain the variation in fruit color, there is still a possibility that a true mutation may have caused these differences. However, traits that are influenced by multiple genes in complex pathways, such as fruit development, pose significant challenges to research. Maturation and ripening of fruits are complex traits that involve various genes in the fruit development process; therefore, the results of mutation studies are diverse. Most studies have reported results by listing numerous differentially expressed genes with varying expression patterns [47,56,57,61,66], or by listing SNPs or Indels at the genomic level [44,72]. Of course, the importance of these studies is evident because the candidate genes identified in these studies will lead to further research.

A good example of a study that has identified the genetic cause of somatic mutation and validated candidate genes concerned tree growth habits in apples [52]. The study used fine mapping to narrow down the trait-associated region and identified a long terminal repeat (LTR) retrotransposon insertion in that region. The candidate genes were then screened using RNA-Seq, and transgenic lines were created using the candidate gene to verify phenotypic differences. In a second example, a 2.8 Mb hemizygous deletion, which is considered the cause of late maturation in somatic mutations, was identified [22]. Candidate genes were narrowed down using RNA-Seq data from fruit development stages, and transgenic lines were created using the candidate gene *MdACT7* for phenotype verification in *Arabidopsis*. As in the above example, we hope that more somatic mutation studies will be conducted in the future, from the proposed genetic causes to the validation of candidate genes.

## 6. Conclusions

This review provides an overview of the factors that induce somatic mutations, methods used to detect them, and molecular mechanisms involved. Furthermore, we present several case studies demonstrating the potential of somatic mutation research to identify new sources of genetic variation and enhance fruit crop variety. The detection and identification of genes and pathways affected by somatic mutations can also improve our understanding of fruit crop biology and enable precise breeding efforts. As somatic mutation research continues to evolve, several areas warrant further investigation. Further research is needed to understand the genetic and environmental factors that influence somatic mutation rates in different fruit crop species. Advances in molecular techniques, such as NGS, have the potential to greatly expand our understanding of somatic mutations in fruit crops. In addition, the development of various bioinformatics tools and the use of multi-omics analysis systems are expected to facilitate a comprehensive understanding of somatic mutations.

In conclusion, somatic mutation research represents a promising avenue for improving fruit crops and expanding our knowledge of fruit crop biology. By continuing to study somatic mutations and developing new methods for their detection and analysis, we can uncover new sources of genetic variation and facilitate more targeted breeding efforts in fruit crops. In the near future, it is hoped that a smart factory designed to induce somatic mutations in fruit trees will be developed. This smart factory is expected to be capable of creating various environmental stresses to induce mutations that exhibit superior quality or resistance to specific diseases, with a non-destructive screening system for somatic mutations. If such a smart factory can achieve these capabilities, it is anticipated to have tremendous value.

## Figures and Tables

**Figure 1 plants-12-01316-f001:**
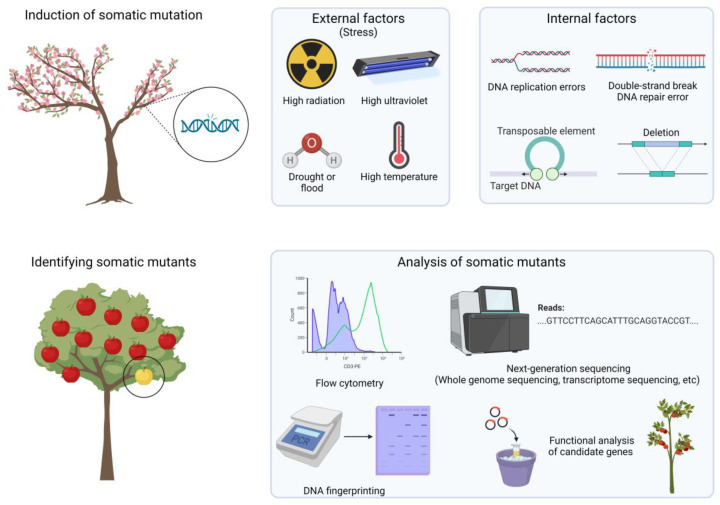
Overview of factors, inducing somatic mutations and methods for analysis in fruit crops.

**Table 1 plants-12-01316-t001:** Molecular techniques for detecting somatic mutation.

Technique	Description	Use
Cytogenetic techniques	Analysis of chromosome number, structure, and behavior.	Detection of chromosomal abnormalities, such as aneuploidy, polyploidy, deletions, and translocations.
Flow cytometry	Measurement of DNA content of individual cells.	Detection of aneuploidy and polyploidy, as well as changes in the DNA content of individual cells.
DNA fingerprinting	Analysis of DNA fragments to identify genetic differences between individuals.	Detection of somatic mutations by comparing the DNA fingerprint of a mutated cell to that of a normal cell.
Microarray analysis	Hybridization of DNA fragments to a chip containing thousands of probes.	Detection of differences in gene expression between mutated and normal cells.
Next-generation sequencing	Rapid generation of large amounts of DNA sequence data.	(1) Comparison of DNA sequence between mutated and normal tissues. (2) Through RNA-Seq, differences in gene expression or isoforms of expressed genes can also be detected.

**Table 2 plants-12-01316-t002:** Case studies of somatic mutations in different fruit crops.

Plant	Mutant Phenotype	Major Findings	Ref.
Apple	Fruit-bearing type (spur)	A novel 2190 bp insertion associated with a preexisting Gypsy-50 retrotransposon was identified in the genome of ‘Red Delicious’ spur mutants using inter-retrotransposon amplified polymorphism (IRAP) and genome walking. The insertion is a spur-specific solo long-terminal repeat (sLTR).	[45]
Apple	Fruit color	The aim of the study was to investigate the genetic basis of the blushed coloring pattern in the fruit skin of a bud sport of *Malus domestica* Borkh. cultivar ‘Ralls’ by comparing it with ‘Ralls’ (striped red). The study found that the DNA methylation in the promoter region of *MdMYB1* played a significant role in regulating *MdMYB1* expression and affecting the color pattern of the apples.	[5]
Apple	Fruit color	A module consisting of 34 genes that were highly correlated with anthocyanin content was identified using RNA-Seq and weighted gene co-expression network analysis (WGCNA). The researchers found that methylation in the promoter region of *MdMYB10* was likely responsible for the yellow color fruit.	[6]
Apple	Fruit color	RNA-seq analysis was used to identify the genetic basis of enhanced coloration in a red ‘Fuji’ apple mutant. A novel *R2R3*-*MYB* transcription factor, *MdMYB90-like*, was discovered to regulate anthocyanin biosynthesis by interacting with other transcription factors. The upregulation *of MdMYB90-like* in the mutant is due to decreased DNA methylation in its promoter region. Transgenic analysis validated that upregulation of *MdMYB90-like* increases the expression of genes associated with anthocyanin production.	[46]
Apple	Fruit maturation date	Using next-generation sequencing, single nucleotide polymorphisms (SNPs) and insertions/deletions (InDels) were detected in the ‘Fuji’ apple and its bud mutant cultivars, and unique genetic variations were identified for each bud mutant.	[44]
Apple	Fruit maturation date	Differential gene expression of ethylene biosynthesis and signaling genes, along with a cell-wall degradation enzyme, was observed between the mutant and its parental variety.	[47]
Apple	Fruit maturation date	Through a combination of RNA-Seq and genomic sequencing, a hemizygous deletion of 2.8 Mb on chromosome 6 was identified in a late-maturing mutant. The 2.8 Mb hemizygous deletion found on chromosome 6 in the late-maturing mutant was replaced by a 10.7 kb retrotransposon insertion from chromosome 7, resulting in the loss of the functional *MdACT7* allele that may be responsible for early fruit maturation.	[22]
Apple	Leaf albinism	This study utilized bisulfite sequencing and RNA sequencing to examine diverse types of albinism in apple seedlings, leading to the discovery of differentially methylated regions (DMRs) and differentially expressed genes. Nine genes involved in the pathways of carotenoid metabolism and flavonoid biosynthesis were found to be associated with the identified DMRs.	[48]
Apple	Parthenocarpy	The apple PI homolog (*MdPI*) was cloned and a retrotransposon insertion causing loss of function was identified. This resulted in parthenocarpy fruit development in apple.	[49]
Apple	Tree-growth habit (columnar growth)	Researchers used classical and NGS analysis to explore the genetic basis of columnar growth in apple trees, discovering a *Ty3/Gypsy* retrotransposon insertion at 18.8 Mb as the only genomic difference between columnar and non-columnar trees. RNA-seq data show that the columnar growth habit in ‘McIntosh’ and ‘McIntosh Wijcik’ is linked to the retrotransposon transcript’s differential expression, which alters the expression of numerous protein-coding genes.	[50]
Apple	Tree-growth habit (columnar growth)	A 1956 bp non-coding DNA element unique to *Pyreae* was identified in the *Co* region of ‘Wijcik’ when compared to its wild-type ‘McIntosh’. Among the candidate genes found in the *Co* region, the *MdCo31* was up-regulated in axillary buds of ‘Wijcik’. Constitutive expression of *MdCo31* in *Arabidopsis thaliana* resulted in plants exhibiting a columnar growth phenotype.	[51]
Apple	Tree-growth habit (columnar growth)	A single dominant gene, *Co*, controls the phenotype that was fine mapped to a 101 kb region. The study found an 8202 bp LTR retroposon insertion mutation in the Co region, which was closely linked to the columnar growth phenotype. The 91071 gene, located near the insertion mutation, was identified as a possible candidate gene responsible for the phenotype. Overexpressing the 91071 gene in transgenic apples produced a similar phenotype to columnar apples.	[52]
Apple	Tree-growth habit (columnar growth)	This study utilized pooled-genomic sequencing between columnar and standard seedlings for a genetic mapping. Two loci of recessive suppressors (*c2* and *c3*) were discovered to be linked to the repression of the *Co* gene expression, which is induced by retroposon and associated with the columnar phenotype in apple trees.	[53]
Citrus	Fruit color	The researchers utilized a citrus microarray to perform transcript profiling and discovered that the mutant exhibited reduced expression levels of a citrus ortholog of STAY-GREEN genes.	[54]
Citrus	Flowering	The researchers stimulated early growth of lateral buds in fruit-bearing shoots and observed that the absence of the repressive H3K27me3 marks of *CcMADS19* locus in old leaves was linked to the phenomenon. Conversely, young leaves still retained the H3K27me3 marks.	[55]
Citrus	Fruit maturation date	This study aimed to compare the sugar and acid content and the expression of metabolic enzymes during fruit ripening of a late-ripening mutant and its parental line. The mutant exhibited delayed expression of citrus sucrose synthase (*CitSS1*) and higher expression of citrus acid invertase (*CitAI*) compared to the parental cultivars.	[56]
Citrus	Fruit ripening	The researchers used two *Citrus clementina* mutants with delayed color break. The study showed that when the *CcGCC1* gene was down-regulated, the color break was delayed due to genetic, developmental, and hormonal factors.	[57]
Grape	Fruit color	The loss of color in white varieties of *Vitis vinifera* is attributed to the insertion of a retrotransposon in the *VvmybA1* gene	[4]
Grape	Fruit color	The researchers discovered that the presence or absence of the red-colored allele at the berry-colored locus is responsible for determining the color.	[58]
Grape	Fruit color	The study unveiled that the recovered color mutant harbors a heterozygous *VvmybA1* locus, consisting of a non-functional *VvmybA1a* allele and a novel *VvmybA1BEN* allele. The presence of *VvmybA1BEN* restored *VvmybA1* transcripts.	[59]
Grape	Fruit color	This study discovered that an SNP mutation in the promoter region of the *VvmybA1* gene caused the color change from red to black. The color difference caused by the SNP was verified by producing red cells through Agrobacterium-mediated transformation.	[60]
Grape	Fruit development and ripening	RNA-seq was used to identify differentially expressed genes between a mutant and its parent, with a focus on genes involved in berry development and ripening.	[61]
Grape	Fruit quality (taste and color)	In this study, RNA sequencing was conducted to compare two cultivars, and a total of 5388 genes were identified to be associated with changes in total soluble solid and anthocyanin contents. Two significant genes, namely 4-coumarate-CoA ligase and copper amine oxidase, were found to play a crucial role in the changes in total soluble solid and anthocyanin levels caused by bud sport.	[62]
Grape	Fruit size	Whole genome resequencing and transcriptomic sequencing were conducted. Genetic variations related to cell death, symbiotic microorganisms, and other processes, as well as differentially expressed genes related to cell-wall modification, stress response, and cell killing, were identified.	[3]
Grape	Seedless	Differential-gene-expression analysis was conducted between a seeded wine grape and its seedless somatic mutant at three developmental stages. A total of 1075 differentially expressed genes were identified, highlighting significant coordination and enrichment of pollen and ovule developmental pathways.	[63]
Grape	Seedless	In this study, researchers used quantitative genetics, fine-mapping, and RNA-sequencing to identify the primary causal factor of seedlessness in grapes as the AGAMOUS-LIKE11 (*VviAGL11*) gene. Specifically, they discovered that a single-point variation in *VviAGL11* resulting in an arginine-197-to-leucine substitution was fully associated with stenospermocarpy.	[64]
Orange	Fruit acidity	A reference genome of sweet orange and six diploid genomes of somatic mutants were assembled de novo. Subsequently, 114 somatic mutants were sequenced, revealing somatic mutations, structural variations, and transposable-element transpositions, including transporter or regulatory gene insertions linked to fruit acidity variation.	[8]
Orange	Fruit quality	By searching for differentially expressed genes using subtractive hybridization and microarray analysis, 13 signal transduction- or transcription-factor genes were identified.	[65]
Peach	Fruit maturation date	Differentially expressed genes (DEGs) identified during fruit development stages using RNA-Seq are associated with carotenoid biosynthesis, starch and sucrose metabolism, plant hormone signal transduction, flavonoid biosynthesis, and photosynthesis.	[66]
Peach	Fruit shape (flat shape)	It was discovered that a deletion of approximately 10 kb was present and affecting a gene that co-segregates with the trait. This gene was identified as being orthologous to leucine-rich repeat receptor-like kinase (*LRR-RLK*).	[67]
Peach	Fruit shape (round shape)	Loss of heterozygosity event was identified by NGS in bud sport as the potential cause for alteration in fruit shape. Furthermore, a genome-wide association study involving 127 peach accessions was performed and a single nucleotide polymorphism associated with variations in fruit shape was identified.	[68]
Pear	Fruit quality (sucrose)	In this study, the expression levels of the SWEET gene, *PuSWEET15*, were compared between a somatic-mutant high-sucrose pear variety (BNG) and its parent (NG) using RNA-Seq data. The study discovered that *PuSWEET15* was expressed at higher levels in BNG; overexpression of this gene in NG increased the sucrose content, whereas silencing it in BNG decreased it. The study also revealed that the WRKY transcription factor *PuWRKY31* was expressed to greater levels in BNG fruit and was found to bind to the *PuSWEET15* promoter and induce its transcription. Additionally, *PuWRKY31* was found to upregulate the transcription of ethylene-biosynthetic genes.	[7,69]
Pear	Fruit skin color	The study found a correlation between hypermethylation of the *PcMYB10* promoter and the green-skin phenotype. Infiltrating red-skin fruits with a plasmid to silence endogenous *PcMYB10* resulted in blocked anthocyanin biosynthesis.	[70]
Pear	Fruit skin color	A high correlation was found between the accumulation of anthocyanin and the expression of genes *PpUFGT2* and *PyMYB10*. The red bud sport of ‘Zaosu’ pear and the striped pigmentation pattern of ‘Zaosu Red’ pear were found to be related to the demethylation of the *PyMYB10* promoter.	[71]
Pear	Large fruit	This study identified 61 core cell-cycle genes by transcriptome analysis.	[2]
Plum	Fruit-ripening type	The genomic DNA of six plum cultivars were sequenced using genomic sequencing. Potential genes related to ethylene perception and signal transduction were identified as potential candidate genes.	[72]

## Data Availability

Not applicable.

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
