# Peer review of "Somatic Mutations in Fruit Trees: Causes, Detection Methods, and Molecular Mechanisms"

_plants, 2023, doi:10.3390/plants12061316_

Round 1

Reviewer 1 Report

Dear Authors for you what is the difference of clones, crosses, mutations , bud spots ?. enlightening ,....

Can you include one paragraph about this theme and discuss this in the article. I think that will be enriching for the paper, and help you to enlightening the plant genetic diversity within a variety.

ln row  37 you write ....asexual reprodution, in plants.... i think that is more correct write ....asexual propagation...

In overall the article is aceptable with minor changes, like suggested above.

Author Response

Thank you very much for giving us the chance to revise our manuscript. We are grateful to the editor and reviewers for their insightful feedback and suggestions on our paper. The changes we made are highlighted in red.

Dear Authors for you what is the difference of clones, crosses, mutations , bud spots ?. enlightening ,....

Can you include one paragraph about this theme and discuss this in the article. I think that will be enriching for the paper, and help you to enlightening the plant genetic diversity within a variety.

Response: Thank you for giving us ideas to make our paper more enriched.  The section from line 49 to 57 in the revised manuscript has been updated with additional information on clone, bud, and cross.

ln row  37 you write ....asexual reprodution, in plants.... i think that is more correct write ....asexual propagation...

Response: I have made the changes as you suggested.

Reviewer 2 Report

The text should be better organise, I would suggest to add to conclusions authors own points how to improve induction and usage of somatic mutation in fruit tree.

Some details:

Lines 27-29: somatic mutation can not exclude further conversión of these cells to reproductive organs through buds and flowers formation. It will be nice also mention that in fruit tree the major way of propagation is vegetative through grafting. So, prop’agation through gametes has only a minor contribution…

Figure 1: I would sugegst to write induction, not occurrence.  Moreover, all externak factors authors mentioned can be renamed as stress. And oxidative dqamage is just a part of the stress. Please, re-adjust this figure.

Moreover, Flow cytometry is not a methods of mutation análisis. It can only detect polyploidy.

Line 70: “continuous exposure of plants to mutagens in the environment, such as ultraviolet radiation” ¿?? Ultaviolet radiation is a normal condition for plant grotwh, indeed!

Line 72: “can influence” – induced??

Line 78: “Modifications in DNA methylation and histones” ¿??  Do you mean histone modification?

Line 89: “during tissue culture” – please, do not confuse somatic mutation in planta with somaclonal variation.

Line 93 – 114: please, make this part more logical, do not confuse gene expressión and identification of mutation etc.

Line 119: “molecular mechanisms responsible for somatic mutations in fruit crops (Table 2)”??

I do not see molecular mechanism responsable for somatic mutation here. It is characterisation of somatic mutation.

Author Response

Thank you very much for giving us the chance to revise our manuscript. We are grateful to the editor and reviewers for their insightful feedback and suggestions on our paper. The changes we made are highlighted in red.

The text should be better organise, I would suggest to add to conclusions authors own points how to improve induction and usage of somatic mutation in fruit tree.

Response: Thank you for giving us ideas to make our paper more enriched. The section from line 230 to 235 in the revised manuscript has been updated with additional information on how to improve induction and usage of somatic mutation in fruit tree.

Lines 27-29: somatic mutation can not exclude further conversión of these cells to reproductive organs through buds and flowers formation. It will be nice also mention that in fruit tree the major way of propagation is vegetative through grafting. So, prop’agation through gametes has only a minor contribution…

Response: Your suggestions have been very helpful in our revised paper. The section from line 30 to 34 in the revised manuscript has been updated. Furthermore, the statement "In particular, grafting is a major propagation method for fruit trees" has been newly added to line 52.

Figure 1: I would sugegst to write induction, not occurrence.  Moreover, all externak factors authors mentioned can be renamed as stress. And oxidative dqamage is just a part of the stress. Please, re-adjust this figure.

Response: We have made the changes as you suggested.

Moreover, Flow cytometry is not a methods of mutation análisis. It can only detect polyploidy.

Response: Flow cytometry has also been included because there are cases where mutant varieties that have changes in ploidy levels result in larger fruit sizes. If the reviewer agrees, I would like to keep it as is.

Line 70: “continuous exposure of plants to mutagens in the environment, such as ultraviolet radiation” ¿?? Ultaviolet radiation is a normal condition for plant grotwh, indeed!

Response: You said it correctly. What we originally meant was a very strong ultraviolet radiation. We have changed it to "strong ultraviolet radiation" to reflect our intended meaning. We apologize for the confusion.

Line 72: “can influence” – induced??

Response: We have made the changes as you suggested.

Line 78: “Modifications in DNA methylation and histones” ¿??  Do you mean histone modification?

Response: The sentence was confusing and unnecessary as the following sentences already provide sufficient explanation. Therefore, we have deleted the sentence.

Line 89: “during tissue culture” – please, do not confuse somatic mutation in planta with somaclonal variation.

Response: As suggested, the tissue culture section has been removed from the manuscript.

Line 93 – 114: please, make this part more logical, do not confuse gene expressión and identification of mutation etc.

Response: The previously confusing sentence has been extensively revised in line 107-137. Thank you.

Line 119: “molecular mechanisms responsible for somatic mutations in fruit crops (Table 2)”??

I do not see molecular mechanism responsable for somatic mutation here. It is characterisation of somatic mutation.

Response: You are correct. The sentence has been revised as you pointed out.

Round 2

Reviewer 2 Report

The paper is well adjusted. Please, make some modification in figure 1.

High radiation, high UV and explanation of water stress. Do you mean drought stress???

Author Response

Thank you very much for giving us the chance to revise our manuscript. We are grateful to the editor and reviewers for their insightful feedback and suggestions on our paper.

The paper is well adjusted. Please, make some modification in figure 1.

High radiation, high UV and explanation of water stress. Do you mean drought stress???

Response: We sincerely appreciate the comment suggesting a more specific modification of "water stress" to either "drought" or "flood".